# Long-term health effects perceived by snakebite patients in rural Sri Lanka: A cohort study

**Subodha Waiddyanatha**[1,2], **Anjana Silva**[1,2,3], **Kosala Weerakoon**[1], **Sisira Siribaddana**[4], **Geoffrey K. Isbister**[2,5]*

**1** Department of Parasitology, Faculty of Medicine and Allied Sciences, Rajarata University of Sri Lanka, Saliyapura, Sri Lanka, **2** South Asian Clinical Toxicology Research Collaboration (SACTRC), Faculty of Medicine, University of Peradeniya, Peradeniya, Sri Lanka, **3** Monash Venom Group, Department of Pharmacology, Faculty of Medicine, Nursing and Health Sciences, Monash University, Australia, **4** Department of Medicine, Faculty of Medicine and Allied Sciences, Rajarata University of Sri Lanka, Saliyapura, Sri Lanka, **5** Clinical Toxicology Research Group, University of Newcastle, Newcastle, New South Wales, Australia

* geoff.isbister@gmail.com

**Data Availability Statement:** All data files are available from the University of Newcastle repository database http://hdl.handle.net/1959.13/ 1433597

## Abstract

The acute effects of snakebite are often emphasized, with less information on long-term effects. We aimed to describe the long-term health effects perceived by patients followed up after confirmed snakebites. Two groups of snakebite patients (>18y) from the Anuradhapura snakebite cohort were reviewed: Group I had a snakebite during August 2013-October 2014 and was reviewed after 4 years, and group II had a snakebite during May 2017-August 2018, and was reviewed after one year. Patients were invited by telephone, by sending letters, or doing home visits, including 199 of 736 patients (27%) discharged alive from group I and 168 of 438 patients (38%) from group II, a total of 367 followed up. Health effects were categorised as musculoskeletal, impact on daily life, and medically unexplained. Health issues were attributed to snakebite in 107/199 patients (54%) from group I and 55/168 patients (33%) from group II, suggesting the proportion with health issues increases with time. Sixteen patients (all viperine bites) had permanent musculoskeletal problems, none with a significant functional disability affecting daily routine. 217/367 reported being more vigilant about snakes while working outdoors, but only 21/367 were using protective footwear at review. Of 275 farmers reviewed, only six (2%) had restricted farming activities due to fear of snakebite, and only one stopped farming. 104/199 (52%) of group I and 42/168 (25%) of group II attributed non-specific symptoms (fatigue, body aches, pain, visual impairment) and/or oral cavity-related symptoms (avulsed teeth, loose teeth, receding gums) to the snakebite, which cannot be explained medically. In multivariate logistic regression, farming, type of snake, antivenom administration, and time since snakebite were associated with medically unexplained symptoms. The latter suggests medically unexplained effects increased with time. Based on two groups of snakebite patients reviewed one and four years post-bite, we show that long-term musculoskeletal disabilities are uncommon and not severe in snakebite survivors in rural Sri Lanka. However, a large portion of patients complain of various non-specific general and oral symptoms, not explainable based on the

**Funding:** This research was funded by Australian National Health and Medical Research Council (NHMRC) through the Centers for Research Excellence grant number 1110343 (https://www.nhmrc.gov.au/), recieved by GKI. The funders had no role in the study design, data collection and analysis, decision to publish, or preparation of the manuscript.

**Competing interests:** The authors have declared that no competing interests exist

known pathophysiology of snakebite. These perceived effects of snakebite were more common in patients with systemic envenoming, and were more frequent the longer the time post-bite.

## Author summary

Snakebite is a significant public health issue in the tropics, and the rural farming communities in developing countries are at the greatest risk of snakebite. The acute effects of snakebite are often emphasized in the literature. There is a severe dearth of prospective studies to obtain reliable data about the long-term impact. We aimed to describe the long-term health effects perceived by patients followed up after confirmed snakebites. We followed up 199 and 168 confirmed snakebite patients from a snakebite cohort in rural Sri Lanka four and one year after snakebite. One-third of the patients after one year and half of the patients after four years reported various long-term effects attributed to the snakebite, suggesting the proportion with health issues increases with time. However, only sixteen patients (all viperine bites) had permanent musculoskeletal problems, none with a significant functional disability affecting daily routine. Many patients complain of various non-specific and oral symptoms that they perceived as effects of the snakebite. Although these symptoms were more common in patients with systemic envenoming and viper bites, they cannot be explained as venom-induced effects, warranting a broader socio-cultural explanation.

## Introduction

Snakebite is a neglected tropical disease that primarily affects rural communities in tropical regions of Asia, Africa and Latin America[1–3]. Literature-based estimates suggest up to 5.5 million snakebites, 1.8 million snake envenomings and 94,000 estimated deaths occur due to snake envenoming in the world annually [1]. Most victims of snakebite are young and middle-aged agricultural workers from less-privileged communities [3]. South Asia records the highest-burden of snake envenoming in the world and over two-thirds of snakebite mortality [4]. Community-based studies estimated 398 snakebites, 151 envenomings per 100,000 population per year occurring in Sri Lanka[5]. Five snakes: Russell's viper (*Daboia russelii*), saw-scaled viper (*Echis carinatus*), Indian krait (*Bungarus caeruleus*), common cobra (*Naja naja*)] and hump-nosed pit viper (Genus: *Hypnale*) are responsible for the majority of medically important snakebites in India and Sri Lanka[4,6].

Snake envenoming can cause well recognised acute effects, including local tissue injury and necrosis, venom-induced consumption coagulopathy (VICC), neuromuscular paralysis, myotoxicity and acute kidney injury [7]. These acute effects are routinely treated in hospitals, and patients receive antivenom and/or supportive care. Our current knowledge of snake envenoming almost entirely focuses on these acute effects, the morbidity and mortality associated with them, and their treatment. Little is known about the long-term effects of snake envenoming [8,9]. This is most likely because once the life-threatening effects of snake envenoming are treated in hospital, snakebite patients are not actively followed up and patients rarely seek further medical advice for any ongoing effects [8]. However, snake envenoming can result in long-term sequelae that are physically or psychologically debilitating. It can compromise the quality of the patient's life and socioeconomically affect their families [8,10–12].

A recent scoping review highlighted the dearth of prospective studies on the long-term effects of snake envenoming [8]. A broad range of long-term outcomes and complications have been reported following snakebite. These vary from objective physical impairments, such as amputations, permanent scarring and eye injury, to far more subjective effects of the snakebite perceived by patients. The latter is problematic because although they may be psychosomatic in nature, they can still significantly impact a patient's well-being and their ability to function in society. Snakebite survivors from Sri Lanka have attributed a range of health effects that are difficult to explain medically, such as 'migraine-like syndromes' and various bodily aches and pains, even years after the snakebite [13,14]. Whether such medically unexplained symptoms are true somatic effects of snakebite or originate from the 'knowledge that they were envenomed' needs exploration. The latter is common in Asian populations and of which specific communities respond to emotional distress in a highly culture-bound manner [15].

Previous studies of the long-term effects have been based on selected patient groups at follow-up, providing limited information on prevalence data and the initial acute envenoming [8]. To better understand the long-term effects of envenoming, prospective cohort studies are required to document acute envenoming with accurate case authentication and active follow-up of patients.

We aimed to describe the long-term effects in snakebite patients followed up at one and four years post-bite from a snakebite cohort in rural Sri Lanka. In addition, we aimed to investigate factors associated with patients developing long-term effects of snakebite.

## Methods

### Ethics statement

Human ethics approval was obtained from the Ethics Review Committee of the Rajarata University of Sri Lanka for prospective patient enrolment (ERC/2012/036, ERC/2013/019) and to review patients (ERC/2017/047). Written informed consent was obtained from all patients on admission and at review. Proxy consent (verbally taken from the guardian) was obtained on admission when the patients were not conscious and severely ill. Written informed consent was then taken before discharge after the patient regained consciousness.

### Anuradhapura snakebite cohort

We prospectively followed up two groups of patients from a snakebite cohort in rural Sri Lanka, recruited at the time of the bite, to investigate the long-term effects of envenoming. The study was undertaken at the Teaching Hospital, Anuradhapura, Sri Lanka. This is the third-largest medical facility in the country and covers the North Central Province of Sri Lanka. The province has a large geographical area with agricultural lands and records the highest incidence of snake envenomings in the country, three times higher than the national incidence [5].

The Anuradhapura Snakebite Cohort prospectively recruits all confirmed snakebite patients aged over 16 years presenting to the study centre and records epidemiological and pre-hospital data, clinical effects, laboratory investigations and treatment details for each case. The patient recruitment for the study was in operation from August 2013 to October 2014 and continues from May 2017. Patient recruitment for Anuradhapura Snakebite Cohort was stopped between November 2014 to April 2017.

For the Anuradhapura snakebite cohort, all recruited patients have a clinical assessment on admission to hospital and daily thereafter. Serial clinical assessments are done at 1, 4, 8, 12 and 24 h post-bite, and then every 24 h until discharge. All clinical assessments are done by medically qualified clinical research assistants. All laboratory investigations and interventions are done as part of patient management and are prospectively recorded. All clinical and laboratory

data are recorded using a pre-formatted clinical data form and are prospectively entered into a relational database. The epidemiological data and the clinical outcomes for patients recruited between August 2013 and October 2014 (Group I) have been previously published [16–20].

Patients enrolled in the Anuradhapura snakebite cohort have the snakebite authenticated if the patient has identifiable fang or teeth marks, features of local or systemic envenoming or the patient witnessed the snakebite. If the snake specimen is available, it is identified by an experienced herpetologist (AS). In the remaining patients of cohort I, species-specific sandwich enzyme-linked immunosorbent assay (ELISA) was carried out on pre-antivenom blood samples to determine the snake species as previously reported [16,17]. In cases in which a pre-antivenom blood sample was not available, ELISA was done on the sample following venom dissociation [18]. In patients of cohort II, the snake was authenticated using a previously validated syndromic approach based on the Sri Lankan venomous snake identification algorithm [21].

## Patients

For this study of long-term effects, patients recruited from August 2013 to October 2014 (group I) to the Anuradhapura snakebite cohort were reviewed four years after the bite, in 2017 and 2018. Patients recruited to the cohort from May 2017 to August 2018 (group II) were reviewed one year after the bite, in 2018 and 2019. All reviews were conducted at the Teaching Hospital, Anuradhapura. The participants were initially invited by telephone and those without contact numbers and unable to be contacted by phone were sent a letter to their residence. In Group II, when not contactable by phone or letter, a research assistant went to the address of the participant to verbally invite them for the hospital review. To encourage participation, an honorarium was provided, which was approved by the Human Ethics Review Committee.

## Data collection

Data on the snakebite was extracted from the database. This included age, sex, snake involved, co-morbidities, clinical details and interventions from the acute admission. Further information was then collected when the patients were followed up. At the time of follow-up, each participant was interviewed by a single physician investigator (SW) for approximately 30 min to obtain a history followed by a complete physical examination.

A formal systematic inquiry about any complications at the bite site and any other musculoskeletal, endocrine, neurological problems was recorded on a pre-prepared case record form (supplementary material). Previously reported medically unexplained effects in snakebite patients [11,13] were recorded, including the presence of excessive hair loss, fatiguability, body aches and pains, photophobia, blurred vision, blackouts, and issues related to the oral cavity using specific pre-defined questions on the data record form. In addition, information was recorded in regards to re-experiencing and flashback of the snakebite, avoidance of routine activities due to fear of snakebite, behavioural changes to avoid snake encounters, and curtailing or quitting farming activities due to the fear of snakebite were specifically asked about.

Persistent effects at the bite site and the affected limb (muscle wasting, chronic wounds, numbness, deformities, and scarring), motor function of all four limbs, endocrine abnormalities (myxoedema, hair loss, secondary sexual characteristics), and effects of chronic kidney disease (skin pigmentations, blood pressure, pallor) were specifically noted. A blood sample and a urine sample were collected to investigate renal function (reported elsewhere).

## Outcomes

The outcomes were categorised as musculoskeletal effects, impact on daily life, and health problems perceived by the patients in relation to snakebite, which cannot be explained based

on the current understanding of the pathophysiology of snake envenoming (medically unexplained symptoms). Medically unexplained symptoms were divided into non-specific symptoms and symptoms related to the oral cavity.

## Data analysis

Continuous data were described using median and interquartile ranges (IQR) and were analysed using non-parametric statistical methods. For comparison of continuous variables without normal distribution, the Mann-Whitney U test was used. For comparisons of proportions of patients for categorical variables, a two-proportion z test was used; $p < 0.05$ was considered significant. A two-step procedure was used to determine the predictors for the presence or absence of any symptom that cannot be explained medically (oral and generalized issues). Step one was a univariate logistic regression to identify potential predictors and crude odds ratios with 95% CI were calculated. Multivariate binary logistic regression analysis was then undertaken with a forward selection approach using the variables that showed statistical significance for a p-value less than 0.05 in the univariate logistic regression analysis. The model produced adjusted odds ratios and 95% CI with the significance level for variables of interest, and the model goodness of fit was evaluated with $R^2$ values for each model. All analyses were done using GraphPad PRISM 9.1 (GraphPad Software, San Diego, CA, USA) and R statistical software version 4.1.0.

## Results

We reviewed 199 patients (27%) of the 736 patients discharged alive from group I after four years and 168 patients (38%) of the 438 patients discharged alive from group II after one year (Fig 1). There were only minor differences between reviewed patients and those not reviewed, with more of the reviewed patients being definite Russell's viper bites, developing envenoming and receiving antivenom (Table 1). In both reviewed groups, the typical snakebite patient was a middle-aged male farmer, with 72% and 69% being confirmed viperine (Russell's viper and Hump-nosed viper) bites, respectively.

At review, 107 of 199 patients (54%) from group I and 55 of 168 patients (33%) from group II had health issues that they attributed to the snakebite.

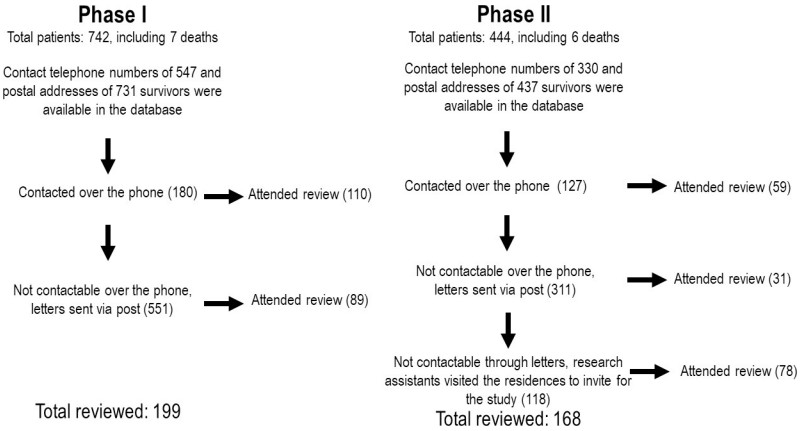

**Fig 1. Patient recruitment for the study.**

**Table 1. The demographic characteristics of all patients and the reviewed patients of group I and group II of the Anuradhapura Snakebite Cohort.**

| | Group I | | Group II | |
|---|---|---|---|---|
| | Non-reviewed patients n = 543 | Reviewed patients n = 199 | Non-reviewed n = 276 | Reviewed patients n = 168 |
| **Age at the snakebite: median (IQR) years** | 38 (26–51) | 44 (32.5–52) | 39 (28–51) | 45 (35–53) |
| **Sex–Male: n, (%)** | 354 (65) | 122 (61) | 178 (64) | 111 (66) |
| **Snake: n, (%)** | | | | |
| Russell's viper | 154 (28) | 93 (47) | 54 (20) | 67 (40) |
| Merrem's Hump-nosed viper | 106 (20) | 49 (25) | 59 (21) | 49 (29) |
| Indian krait | 21 (4) | 12 (6) | 7 (3) | 5 (3) |
| Common cobra | 10 (2) | 0 (0) | 5 (2) | 2 (1) |
| Identified non/mildly venomous snakes | 25 (5) | 15 (7) | 27 (10) | 19 (11) |
| Snake unidentified | 227 (48) | 30 (15) | 124 (45) | 26 (13) |
| **Highest education level: n, (%)** | | | | |
| ≥ Grade 10 in school | 223 (41) | 90 (45) | 121 (44) | 86 (51) |
| **Full or part-time farmers (%)** | 354 (65) | 144 (73) | 173 (62) | 131 (78) |
| **Site of bite: n, (%)** | | | | |
| Lower limb: ankle and foot | 420 (77) | 149 (75) | 198 (71) | 117 (70) |
| Lower limb: above the ankle | 39 (7) | 17 (9) | 11 (4) | 5 (3) |
| Upper limb: hand and wrist | 56 (10) | 19 (10) | 42 (15) | 31 (18) |
| Upper limb: arm or forearm | 11 (2) | 7 (3) | 9 (3) | 5 (3) |
| Other | 17 (3) | 7 (3) | 16 (6) | 10 (6) |
| **Length of hospital stay: median (IQR) days** | 2 (1–3) | 2 (1–3) | 2 (1–3) | 2 (1–3) |
| **Clinical effects during the acute stage: n, (%)** | | | | |
| Local envenoming | 372 (68) | 150 (75) | 165 (60) | 115 (68) |
| Blistering, necrosis, gangrene | 51 (9) | 18 (10) | 20 (7) | 18 (11) |
| Amputations | 0 (0) | 1 (0.4) | 0 (0) | 0 (0) |
| VICC | 182 (34) | 73 (37) | 56 (20) | 55 (33) |
| Neurotoxicity | 120 (22) | 68 (34) | 44 (16) | 49 (29) |
| Mechanical ventilation | 18 (34) | 7 (3) | 8 (2) | 2 (1) |
| Acute Renal Failure | 18 (3) | 12 (6) | 14 (5) | 9 (5) |
| AKI requiring dialysis | 4 (1) | 3 (1) | 1 (1) | 2 (1) |
| Deaths | 7 (1) | N/A | 6 (1) | N/A |
| **Received antivenom: n, (%)** | 179 (33) | 87 (43) | 69 (25) | 63 (35) |
| **Acute adverse reactions to antivenom: n, (%)** | 73 (40) | 44 (51) | 23 (33) | 30 (47) |

## Musculoskeletal effects

Four patients from group I (2%)) and 12 patients from group II (7%)) had permanent musculoskeletal complications from the snakebite (Figs 2 and 3). Five of the 16 were Russell's viper bites (two contractures, three with scar formation and three amputations), and 11 were Merrem's hump-nosed viper bites (two contractures, nine with scar formation and one amputation; Figs 2 and 3 and S1 Table). None had a significant functional disability that affected their daily routine or chronic wounds. A further 31 (16%) patients from group I and 20 (12%) from group II complained of mild local effects, including intermittent mild pain (18/367), tingling sensation (8/367) or numbness (11/367) and swelling (9/367) at the bite site at the time of review (S2 Table). In addition, 13 (7%) patients of group I and eight (5%) patients of group II complained that at least one of these effects lasted more than one month but had resolved by the time of review.

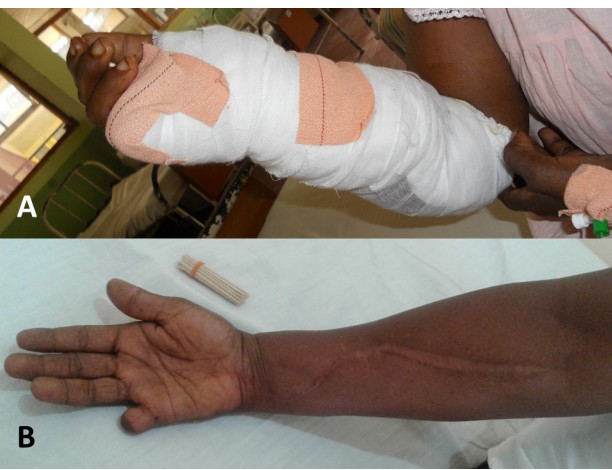

**Fig 2.** A: Sixty-three-year-old female presented with a hump-nosed viper bite to the right small finger and developed severe local necrosis of the distal phalanx and compartment syndrome involving the forearm. She was managed by decompression fasciotomy, followed by amputation of the two distal phalanxes of the right small finger. B: The same patient after four years with lost phalanxes and fasciotomy scar.

## Impact of the bite on daily life

On review, 134/199 (67%) from group I and 83/168 (49%) from group II reported they are constantly alert for snakes while working outdoors, following their snakebite. Fifteen patients (7.5%) from group I and six patients (3.6%) from group II who had not previously worn protective footwear during farming activities had started and continued using protective footwear after the bite. Four patients from group I reported that they avoided the circumstances in which the snakebite occurred.

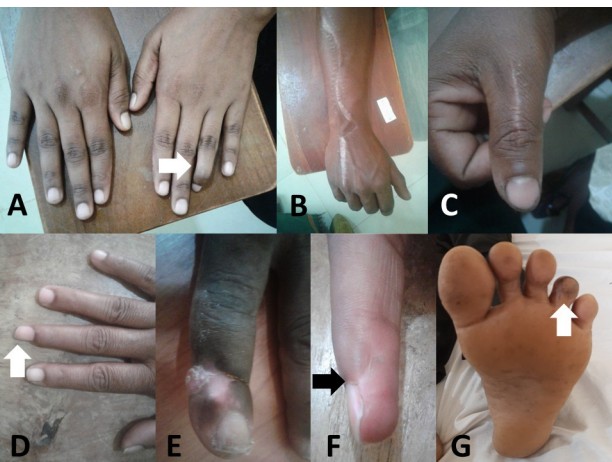

**Fig 3.** **A**: Twenty-six-year-old female with contracture formation in palmer aspect of left ring finger following hump nosed viper bite four years previously. **B**: Twenty-seven-year-old male who developed severe local swelling and compartment syndrome following hump nosed viper bite, one year back, presented with fasciotomy scar extending from the base of the right ring finger up to the elbow. No functional impairment. **C**: Thirty-three-year-old female with a fasciotomy scar on the right thumb following hump nosed viper bite one year back. **D**: Forty-nine-year-old female with tissue loss and scarring in the distal phalanx of the right middle finger following Russell's viper bite one year back. **E**: Forty-six-year-old female with contracture formation in the distal interphalangeal joint of left index finger following Russell's viper bite four years back. **F**: Forty-nine-year-old female with slight tissue loss at the distal phalanx of the right middle finger following Russell's viper bite one year back. **G**: Fifty-eight-year-old male with contracture formation of the plantar aspect of left third toe following hump nosed viper bite one year back.

**Table 2. Effect of the snakebite on the farming activities for the patients.**

| Age (years) | Sex | Farming involvement before snakebite | Snake | Circumstance of snakebite | Acute effects of snakebite | Effect of snakebite on farming activities |
|---|---|---|---|---|---|---|
| 51 | Male | Full-time cattle dairy and paddy farming. Owned 80 cattle before snakebite. | Russell's viper | Bite occurred while grazing cattle. | Developed VICC, neurotoxicity, rhabdomyolysis and mild local effects, and non-specific systemic effects. Developed anaphylaxis following the 20 vials of antivenom therapy. Hospitalised for 5 days. | Sold all the cattle due to the fear of snakebite. After two years, the patient had started cattle grazing again. Paddy farming was never stopped. The patient now regrets selling his herd. |
| 54 | Female | Full-time cattle grazing and paddy and chena farming. Owned 25 cows. | Russell's viper | Bite occurred while grazing cattle. | Developed VICC, neurotoxicity and mild local effects, and non-specific systemic effects. 20 vials of antivenom were given, without reactions. Hospitalised for four days. | Sold all cows due to the fear of snakebite. Gave up cattle grazing, still involved in paddy and chena farming. |
| 63 | Female | Full-time paddy and chena farming. | Russell's viper | Bite occurred while manually harvesting paddy. | Developed cardiovascular collapse and resuscitated VICC, neurotoxicity and mild local effects, and non-specific systemic effects. The patient was mechanically ventilated for one day following the cardiovascular collapse. 20 vials of antivenom were given without reactions. Hospitalised for 4 days. | Does not engage in paddy harvesting activities. However, continues other farming activities. |
| 57 | Female | Full-time paddy and chena farming | Russell's viper | Bite occurred while land preparation for chena. | Developed VICC, acute renal failure, and mild local effects, and non-specific systemic effects. Developed anaphylaxis following the 20 vials of antivenom therapy. Hospitalised for 8 days. | Stopped chena farming, however, continues paddy farming. |
| 45 | Male | Full-time paddy and chena farming | Russell's viper | Bite occurred while harvesting paddy using a manual paddy reaper. | Developed VICC, neurotoxicity and mild local effects, and non-specific systemic effects. 20 vials of antivenom were given, without reactions. Hospitalised for three days. | Given up all farming activities and works as a mechanic. |
| 51 | Male | Full-time paddy and chena farming | Russell's viper | While irrigating the paddy field. | Developed hypotension, VICC, neurotoxicity, and moderate local effects. 40 vials of antivenom were given, without reactions. Hospitalised for nine days. | Stopped paddy farming. Continues chena farming and works as a manual labourer. |

**Impact on farming.** There were 144 full-time or part-time farmers reviewed from group I. Only six (4%) had restricted their farming activities due to the fear of another snakebite. Only one patient had completely stopped farming activities and switched to another occupation. One patient from group I had stopped cattle farming completely post-bite but re-started two years later and regretted that he sold the cattle after the snakebite (Table 2). None of the 131 full-time or part-time farmers from group II had restricted or quit farming activities on review one year after the bite due to the snakebite.

**Psychological effects.** Only one patient from both groups experienced flashbacks following the snakebite. This was a 41-year-old male ex-army soldier from group I, who was bitten by an Indian krait and developed severe neuromuscular paralysis treated with antivenom. He then complained of re-experiencing the bite and the events immediately before he developed respiratory paralysis for the four years after the bite. The patient had re-experienced a feeling of a snake slithering under the bed cover on several occasions during the past four years. On three occasions while sleeping, he felt like choking and difficulty in breathing that he had experienced immediately before intubation. The re-experience events did not impact his daily life. The patient was assessed using the post-traumatic stress disorder (PTSD) checklist (civilian version) validated in Sinhala language and scored 25 (cut-off value for a provisional diagnosis of PTSD is 33). However, the patient was referred to a specialist psychiatrist for further

**Table 3. Non-specific issues that could not be explained medically, complained by reviewed patients.**

| | | Russell's viper (n₁ = 93) (n₂ = 67) | Merrem's Hump-nosed viper (n₁ = 49) (n₂ = 49) | Indian krait (n₁ = 12) (n₂ = 5) | Common cobra (n₁ = 0) (n₂ = 2) | Identified non/mildly venomous snakes (n₁ = 15) (n₂ = 19) | Snake unidentified (n₁ = 30) (n₂ = 26) | All (n₁ = 199) (n₂ = 168) |
|---|---|---|---|---|---|---|---|---|
| *Non-specific* | | | | | | | | |
| Fatigue: n, (%) | Group I | 37 (40%) | 13 (27%) | 2 (17%) | 0 | 1 (7%) | 2 (7%) | 55 (27%) |
| | Group II | 12 (18%) | 2 (4%) | 2 (40%) | 0 | 0 | 1 (4%) | 17 (10%) |
| Body aches and pains: n, (%) | Group I | 17 (18%) | 13 (27%) | 2 (17%) | 0 | 1 (7%) | 2 (7%) | 35 (18%) |
| | Group II | 3 (4%) | 0 | 0 | 0 | 0 | 0 | 3 (2%) |
| Loss of appetite: n, (%) | Group I | 7 (8%) | 4 (8%) | 1 (8%) | 0 | 1 (7%) | 0 | 13 (7%) |
| | Group II | 1 (1%) | 0 | 0 | 0 | 0 | 0 | 1 (1%) |
| 'Aura-like experience': n, (%) | Group I | 7 (8%) | 2 (4%) | 0 | 0 | 0 | 0 | 9 (5%) |
| | Group II | 2 (3%) | 0 | 0 | 0 | 0 | 0 | 2 (1%) |
| Gray hair: n, (%) | Group I | 2 (3%) | 1 (2%) | 0 | 0 | 0 | 0 | 3 (2%) |
| | Group II | 3 (4%) | 0 | 0 | 0 | 0 | 1(4%) | 4 (2%) |
| Scalp hair loss: n, (%) | Group I | 1 (1%) | 0 | 0 | 0 | 1 (7%) | 0 | 2 (1%) |
| | Group II | 0 | 0 | 0 | 0 | 0 | 0 | 0 |

n1, number of patients in group I; n2, number of patients in group II

assessment. A clinical diagnosis of PTSD was not made, and it was decided that there was no indication for pharmacological management.

## Health problems that cannot be explained medically

One hundred and four patients (52%) from group I and 42 patients (25%) from group II complained of symptoms or effects that they attributed to the snakebite, which cannot be reasonably explained medically. These symptoms were broadly categorised as non-specific symptoms (Table 3) and symptoms related to the oral cavity (Table 4). In group I, 95/199 (48%) patients

**Table 4. Patients complaints related to the oral cavity that could not be explained medically.**

| | | Russell's viper (n₁ = 93) (n₂ = 67) | Merrem's Hump-nosed viper (n₁ = 49) (n₂ = 49) | Indian krait (n₁ = 12) (n₂ = 5) | Common cobra (n₁ = 0) (n₂ = 2) | Identified non/mildly venomous snakes (n₁ = 15) (n₂ = 19) | Snake unidentified (n₁ = 30) (n₂ = 26) | All (n₁ = 199) (n₂ = 168) |
|---|---|---|---|---|---|---|---|---|
| Teeth loosening: n, (%) | Group I | 9 (10%) | 6 (12%) | 0 | 0 | 0 | 2 (7%) | 17 (9%) |
| | Group II | 5 (7%) | 1 (2%) | 0 | 0 | 0 | 0 | 6 (4%) |
| Avulsed teeth: n, (%) | Group I | 10 (11%) | 4 (8%) | 0 | 0 | 0 | 3 | 17 (9%) |
| | Group II | 5 (7%) | 0 | 0 | 0 | 0 | 0 | 5 (3%) |
| Brittle teeth: n, (%) | Group I | 8 (9%) | 5 (10%) | 0 | 0 | 0 | 1 (3%) | 14 (7%) |
| | Group II | 0 | 0 | 0 | 0 | 0 | 0 | 0 |
| Receding gums: n, (%) | Group I | 3 (3%) | 2 (4%) | 0 | 0 | 0 | 0 | 5 (3%) |
| | Group II | 0 | 0 | 0 | 0 | 0 | 0 | 0 |
| Teeth sensitivity: (%) | Group I | 2 (2%) | 0 | 0 | 0 | 0 | 0 | 2 (1)% |
| | Group II | 3 (4%) | 0 | 0 | 0 | 0 | 0 | 3 (2%) |

n1, number of patients in group I; n2, number of patients in group II

had non-specific symptoms, and 46/199 (22%) had symptoms related to the oral cavity when reviewed four years after the snakebite. In group II, one year after the snake bite, there were 37/168 (22%) patients with non-specific symptoms and 14/168 (8%) with symptoms related to the oral cavity.

The most frequent medically unexplained non-specific symptom was fatigue, reported by 72/367 (20%) of reviewed patients, followed by new-onset non-specific body aches and pains, following routine daily activities in 38/367 (10%) (Table 3). Fourteen patients complained of loss-of appetite following the snakebite. Eleven of 367 patients (3%) complained of experiencing either photophobia and/or blurred vision and/or blackouts while working under bright sunlight (an aura-like experience), which impaired daily activities after the bite. These aura-like symptoms occurred intermittently and only impaired but did not prevent routine daily activities. Seven patients in group I and seven patients in group II complained of visual impairment (poor vision) following the snakebite. This was after excluding patients who developed cataracts or diabetic retinopathy and referred to specialist ophthalmological care. All these patients attributed poor vision to the snakebite. Nine of the 367 patients attributed scalp hair turning grey and loss of scalp hair to the snakebite.

Forty-six patients of group I and 14 patients of group II reported symptoms related to the oral cavity that they attributed to the snakebite (Table 4). In both groups, the most frequent complaint was loosening of teeth with no history of trauma (23/367). Twenty two of 367 complained of avulsed teeth without trauma, a median of 4 teeth (range 1–10) in group I and 3 teeth (range: 3–5) in group II. In addition, brittle teeth (14/367), receding gums (5/367) and sensitive teeth (5/367) were reported by the patients. There was no difference in age or betel nut chewing between patients who reported oral issues and those who didn't for both groups.

None of the patients in either group reported specific symptoms: cold intolerance, dry skin, alteration of bowel habits, menstrual disturbances, sleepiness following the snakebite. Further, none of the patients reported urinary or sexual problems experienced after the snakebite. Patients did not report neurological effects, including dysfunction in balance, coordination or muscle weakness (in addition to what is reported under musculoskeletal issues).

Univariate logistic regression found that in the combined group (Group I and II), the presence of medically unexplained general and/or oral symptoms was associated with time from the snakebite, farming, viper bites, local envenoming, systemic envenoming, neurotoxicity, coagulopathy, administration of antivenom, anaphylaxis to antivenom and the number of days hospitalised at the time of the bite (Table 5). In further multivariate binary logistic regression analysis in the total cohort (group I and II combined), farming, being bitten by a viper, administration of antivenom, and time from the snakebite were found to be significant predictors among all ten variables considered. The final model had an $R^2$ value explaining 26% of the variance of medically unexplained symptoms, by the predictor variables (Table 6). This suggests that there is an increase in the reporting of symptoms the longer the period after the bite.

## Discussion

We found that following a snakebite, one-third of patients after one year (Group II) and half of the patients after four years (Group I) continued to have health issues that they attributed to the snakebite. Of the patients who had health issues, 97% from group I and 76% in group II complained of one or more symptoms that could not be explained medically, including non-specific symptoms and symptoms related to the oral cavity. A few patients had musculoskeletal complications, including minor amputations, and only 12% and 16% of the patients experienced minor local effects such as recurring pain, numbness, tingling sensation, and swelling at the site of the bite, one year and four years after the snakebite respectively (Table 2). These

**Table 5. Univariable logistic regression analysis: crude odds ratios for associations between symptoms that cannot be explained medically and all exposure variable.**

| | Exposure variable | Outcome | Non specific symptoms present (% for raw total) | | Crude OR | OR 95% CI | Significance (p) |
|---|---|---|---|---|---|---|---|
| | | | Negative | Positive | | | |
| 1 | Age <60 years | Negative | 181 (59.15) | 125 (40.85) | | | 0.35 |
| | | Positive | 40 (13.07) | 21 (86.93) | 0.76 | 0.43 to 1.35 | |
| 2 | Sex | Female | 75 (55.97) | 59 (44.03) | | | 0.21 |
| | | Male | 146 (62.66) | 87 (37.34) | 0.76 | 0.49 to 1.17 | |
| 3 | Time since snakebite | 1 year | 126 (75.00) | 42 (25.00) | | | 0.00 |
| | | 4 years | 95 (47.74) | 104 (52.26) | 0.30 | 0.19 to 0.48 | |
| 4 | Farming | Negative | 70 (72.16) | 27 (27.84) | | | 0.00 |
| | | Positive | 147 (55.47) | 118 (44.53) | 2.08 | 1.25 to 3.45 | |
| 5 | Level of formal education | ≥ O/L | 108 (56.54) | 83 (43.46) | | | 0.15 |
| | | <O/L | 112 (64.00) | 63 (36.00) | 0.73 | 0.48 to 1.11 | |
| 6 | Diabetes mellitus | Negative | 205 (61.01) | 131 (38.99) | | | 0.31 |
| | | Positive | 16 (51.61) | 15 (48.39) | 1.47 | 0.70 to 3.07 | |
| 7 | Hypertension | Negative | 172 (60.56) | 112 (39.44) | | | 0.90 |
| | | Positive | 49 (59.76) | 33 (40.24) | 1.03 | 0.63 to 1.71 | |
| 8 | Pre-existing chronic kidney disease | Negative | 216 (60.50) | 141 (39.50) | | | 0.51 |
| | | Positive | 5 (50.00) | 5 (50.00) | 1.53 | 0.44 to 5.39 | |
| 9 | Type of snake | Non-viperid | 90 (82.57) | 19 (17.43) | | | 0.00 |
| | | Viperid | 131 (50.78) | 127 (49.22) | 4.59 | 2.64 to 7.97 | |
| 10 | Local envenomation | Negative | 80 (78.43) | 22 (21.57) | | | 0.00 |
| | | Positive | 141 (53.21) | 124 (46.79) | 3.20 | 1.88 to 5.43 | |
| 11 | Coagulopathy | Negative | 164 (68.05) | 77 (31.95) | | | 0.00 |
| | | Positive | 57 (45.24) | 69 (54.76) | 2.58 | 1.66 to 4.02 | |
| 12 | Neurotoxicity | Negative | 164 (65.60) | 86 (34.40) | | | 0.00 |
| | | Positive | 57 (48.72) | 60 (51.28) | 2.01 | 1.28 to 3.14 | |
| 13 | Acute kidney injury | Negative | 212 (61.27) | 134 (38.73) | | | 0.10 |
| | | Positive | 9 (42.86) | 12 (57.14) | 2.11 | 0.87 to 5.14 | |
| 14 | Systemic envenomation | Negative | 145 (70.73) | 60 (29.27) | | | 0.00 |
| | | Positive | 76 (46.91) | 86 (53.09) | 2.73 | 1.78 to 4.21 | |
| 15 | AVS administered | Negative | 151 (69.59) | 66 (30.41) | | | 0.00 |
| | | Positive | 70 (46.67) | 80 (53.33) | 2.61 | 1.70 to 4.03 | |
| 16 | Anaphylaxis due to AVS | Negative | 202 (62.73) | 120 (37.27) | | | 0.01 |
| | | Positive | 19 (42.22) | 26 (57.78) | 2.30 | 1.22 to 4.34 | |
| 17 | Betel chewing | Negative | 156 (63.67) | 89 (36.33) | | | 0.06 |
| | | Positive | 65 (53.28) | 57 (46.72) | 1.54 | 0.99 to 2.39 | |
| 18 | Alcohol consumption | Negative | 164 (62.36) | 99 (37.64) | | | 0.18 |
| | | Positive | 57 (54.81) | 47 (45.19) | 1.37 | 0.86 to 2.16 | |
| 19 | Smoking | Negative | 182 (62.54) | 109 (37.46) | | | 0.08 |
| | | Positive | 39 (51.32) | 37 (48.68) | 1.58 | 0.95 to 2.63 | |
| 20 | No. of days hospitalized | ≥ 5 days | 211 (62.43) | 127 (37.57) | | | 0.00 |
| | | <5 days | 10 (34.48) | 19 (65.52) | 3.16 | 1.42 to 7.00 | |
| 21 | Intubation | Negative | 214 (60.28) | 141 (39.72) | | | 0.89 |
| | | Positive | 7 (58.33) | 5 (41.67) | 1.08 | 0.34 to 3.48 | |

effects can be explained as a result of local infection and inflammation, and did not increase in frequency from 1 to 4 years post-bite. Although two-thirds of the patients in group I and half of the patients in group II were constantly alert for potential snakebite, less than 6% had

**Table 6. Multivariable logistic regression analysis associations between non-specific symptoms and exposure variables.**

| Variable | B | SE(B) | Wald | p value | Adjusted odds ratio (95% CI) |
|---|---|---|---|---|---|
| Involved in farming | 0.686 | 0.279 | 6.023 | 0.014 | 1.99 (1.15–3.43) |
| Type of snake | 1.378 | 0.302 | 20.774 | 0.000 | 3.97 (2.19–7.18) |
| AVS administered | 0.634 | 0.247 | 6.608 | 0.010 | 1.89 (1.16–3.06) |
| Time since snake bite | -1.281 | 0.247 | 26.856 | 0.000 | 0.28 (0.17–0.45) |
| MLR equation (Nagelkerke's $R^2$, 0.26) | Logit(p); (-1.677) + 0.686(Involved in farming) + 1.378(Type of snake, viperid) + 0.634(AVS administered) - 1.281 (Time since snake bite, 1 year) | | | | |

B, Beta co-efficient; SE(B), Beta coefficient standard error; CI, Confidence interval

behavioural changes, such as wearing footwear while farming. Only six of all reviewed patients had restricted farming activities, with one quitting farming.

The small proportion of patients with long-term local musculoskeletal disabilities and their severity in our study is consistent with a previous community-based study in the Eastern province of Sri Lanka, which reported musculoskeletal problems in 3.2% of snakebite patients [22]. This is most likely because most venomous bites were from Russell's vipers and hump-nosed vipers, which are not frequently associated with extensive tissue necrosis. However, 11/16 patients with musculoskeletal sequelae were hump-nosed viper victims, suggesting the medical importance of these snakes in terms of long-term effects. The proportion of patients with musculoskeletal effects did not increase over time, compared to the significant increase in reports of unexplained medical effects.

A large proportion of patients in both groups reported symptoms related to their general health (Table 3). Considering the diversity and nature of these effects, it is difficult to determine if there is any causal relationship between these and direct venom effects, based on the current understanding of the pathophysiology of envenoming. This is the reason we have used the term 'medically unexplained symptoms'. Irrespective of whether the effects were venom mediated, the patients still attributed these health issues to the snakebite, making them still important in terms of the patient's ongoing well-being and the impact that the snakebite had on their life. These symptoms were associated with systemic envenoming in both groups and were more common in farmers and those bitten by vipers. Medically unexplained oral issues were not related to common causes in the studied community, such as age or betel chewing.

The multivariate logistic regression model demonstrated that antivenom administration, type of snake (a viperine bite or not), involvement in farming and the time since bite (4 years as opposed to 1 year) were significantly associated with medically unexplained symptoms. However, this only explained 26% of the occurrence of medically unexplained symptoms. This study was not designed to differentiate whether the perceived symptoms truly originated from the direct effects of snake envenoming (venom effects) or were a result of the trauma or stress of 'the experience of getting bitten or envenomed by a snake'. Interestingly, the proportion of patients developing these symptoms doubled in the group followed up after four years, compared to those after one year, and time from the bite was statistically associated with symptoms. This may be because people, over time, attribute symptoms they perceive to a significant event, such as the snakebite. Similar complaints by patients after snakebite has been reported in a case-control study done in Sri Lanka on the delayed psychological morbidity following envenoming. A qualitative arm based on focussed group discussions briefly described the patients' perceived long-term issues related to snakebite under five main themes, "poor vision, tooth decay, body aches, headaches, weakness and tiredness of the body" [23]. A community-based study done in the eastern province of Sri Lanka on the long-term effects of snakebite,

'migraine-like syndrome' which includes episodic headaches, vertigo, and photosensitivity, was reported in addition to other symptoms such as body aches and pains, visual problems, and dental problems. The 'migraine-like syndrome' was the most reported long-term effect of snakebite, in 46 of 816 participants.

Of the reviewed patients in both groups, three quarters were involved in farming. However, the experience of snakebite had little impact on their farming practices. Most stated that they were more watchful of snakes outdoors, but only a few adopted definitive preventive strategies such as wearing protective footwear while engaging in farming and gardening activities. Even less restricted their farming activities, and only one had stopped farming entirely. This may be interpreted as the events of snakebite have become a part of the life of rural farmers and, for them, snakebite itself is not strong enough as an event that could force them to give up farming, which is their primary source of income. These findings differ from a previous report from Sri Lanka that snakebite led to 10% of envenomed snakebite victims that received antivenom stopping their work due to the negative impact of the snakebite [11]. The difference in the above study may be partly explained by the fact that all participants had envenoming and received antivenom, unlike in our study, which included both envenomed and non-envenomed patients.

There are several limitations to our study. Despite using multiple approaches to follow up patients, including inviting the participants with letters, telephone calls and home visits, we were only able to review 27% of patients from group I and 38% from group II. This is most likely because patients who are well or do not have any medical complaints, do not want to be followed up or attend hospital, particularly if it will interfere with their farming activities or daily livelihood. This may mean that patients with complaints that they attribute to the snakebite are over-represented in the reviewed patients, a selection bias. There was also a disproportionately larger number of patients bitten by Russell's vipers included in the study, compared to patients not followed up for both groups I and II. This meant that those who had systemic envenoming and received antivenom were over-represented, introducing a further selection bias. Although the study was largely descriptive and did not aim to compare the health issues between the two groups, there were differences between the groups. The method of snake authentication was also different between groups I and II, also potentially introducing a bias. Our study was intentionally focused on reporting the perceived health issues of the patients to avoid missing any health problems not explained by the traditional bio-medical model. However, this is associated with the subjectivity of self-reporting and therefore introduces information bias.

Since the intention of this study was solely to document the long-term health problems perceived by snakebite patients, the present study lacked a qualitative component to probe patients' explanatory model about the medically unexplained symptoms. Also, whether they are influenced by beliefs in their society towards snakebite. Is there an alternative narrative to the bio-medical model? Qualitative approaches can answer such questions, covering psychological and ethnographical aspects. It is noteworthy that these symptoms require more robust socio-cultural explanations beyond being captured and interpreted by depressive symptom scores developed and validated in different socio-cultural settings as symptoms of somatisation [11].

## Conclusion

One-third of reviewed patients after one year and half of the reviewed patients after four years since a snakebite in our cohort continue to attribute symptoms they perceive to the bite. Long-term musculoskeletal disabilities due to snakebite were uncommon and not severe. Most of

the long-term effects perceived by patients, both non-specific and related to the oral cavity, cannot be explained attributed directly to venom effects of snake envenoming. We emphasize the need for robust exploration of the socio-cultural elements behind these unexplained symptoms through well-designed qualitative studies.

## Supporting information

**S1 Table. Descriptions of the patients with musculoskeletal sequelae following snakebite and their impact.**
(DOCX)

**S2 Table. Local effects persistent in reviewed patients at the review.**
(DOCX)

## Acknowledgments

We thank Umesh Chathuranga and Arunasiri Muhandiram (South Asian Clinical Toxicology Research Collaboration) for their assistance in data collection.

## Author Contributions

**Conceptualization:** Subodha Waiddyanatha, Anjana Silva, Sisira Siribaddana, Geoffrey K. Isbister.

**Data curation:** Subodha Waiddyanatha, Anjana Silva, Sisira Siribaddana.

**Formal analysis:** Subodha Waiddyanatha, Anjana Silva, Kosala Weerakoon.

**Funding acquisition:** Geoffrey K. Isbister.

**Investigation:** Subodha Waiddyanatha.

**Methodology:** Subodha Waiddyanatha, Anjana Silva, Sisira Siribaddana, Geoffrey K. Isbister.

**Project administration:** Anjana Silva.

**Resources:** Geoffrey K. Isbister.

**Supervision:** Geoffrey K. Isbister.

**Validation:** Anjana Silva, Geoffrey K. Isbister.

**Writing – original draft:** Subodha Waiddyanatha.

**Writing – review & editing:** Anjana Silva, Kosala Weerakoon, Sisira Siribaddana, Geoffrey K. Isbister.

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
