## [Decision Letter · Decision Letter 0]

15 Apr 2022

Dear Prof Isbister,

Thank you very much for submitting your manuscript "Long-term health effects perceived by snakebite patients in rural 2 Sri Lanka: a cohort study" for consideration at PLOS Neglected Tropical Diseases. As with all papers reviewed by the journal, your manuscript was reviewed by members of the editorial board and by several independent reviewers. In light of the reviews (below this email), we would like to invite the resubmission of a significantly-revised version that takes into account the reviewers' comments. 

The manuscript requires major revisions (see reviews).

We cannot make any decision about publication until we have seen the revised manuscript and your response to the reviewers' comments. Your revised manuscript is also likely to be sent to reviewers for further evaluation.

Sincerely,

Thomas Junghanss

Guest Editor

Joerg Blessmann

Deputy Editor

The manuscript requires major revisions.

ABSTRACT

Reviewer #2: 

• There are many methodological limitations of this study, and none are mentioned in the abstract. For example, the participation rates were relatively low, the outcome is based on “perception” which may be subject to information bias, and no relative measures of association are reported in the abstract.

Reviewer #3: 

The description is unclear. was group1 recruited 2013-2014 or was this the date of follow-up, i.e. the recruitment was 4 years earlier. same question regarding group 2. It becomes clear only after reading the full text.

METHODS

Reviewer #1: 

The methodology is generally correct, adapted to the purposes of the study which is mainly de-scriptive.

Reviewer #2: 

• Lines 142-151: It is unclear whether the patients in group I had a different authentication method than patients in group II. If so, this can result in both selection bias and information bias and must be acknowledged and addressed by the authors.

• The selection of patients in 2013/2014 vs. 2017/2018 can result in selection bias. I understand that the one year vs. four year follow-up is evaluated. However, differential selection of participants based on different years of study entry may introduce factors in one group (vs. the other) that are not present or which may vary with the other group. For example, did treatment in 2013/2014 differ between 2017/2018? Did access to care differ during those time periods?

• Lines 200-202: It appears as though you are conflating multicollinearity with confounding. These are two different concepts. Multicollinearity occurs when the covariates are highly correlated with each other (potentially resulting in unstable risk estimates). Confounding factors are related independently to the exposure of interest and the outcome of interest.

Reviewer #3: 

Generally, it is fine to perform a logistic regression. I wonder why the authors used separate logistic models for Group 1 and 2. It would be useful to have a joint model, with the group (resp. the time after snake bite) as a covariable. This would also give an adjusted estimate to show a possible difference between one and four years after snake bite. The Fisher test at the beginning of the results section would then become redundant. 

A dichotomization of continuous covariables (here: age) is generally not recommended (see. for example, Patrick Royston, Douglas G Altman, Willi Sauerbrei. Dichotomizing continuous predictors in multiple regression: a bad idea. Stat Med . 2006 Jan 15;25(1):127-41. )

RESULTS

Reviewer #1: 

The described situation is accurately reported, which is an essential starting point for future dis-cussions. It confirms other studies, particularly in Africa, which had shown, on the one hand the low frequency of physical sequelae and on the other hand the importance of psychological disor-ders following snakebites.

Reviewer #2: 

• Lines 208-209: the participation rates were low. This should be acknowledged in the discussion section [you do this but do not discuss the statistical implications of your results]. Furthermore, Table 1 should separate patients reviewed by not reviewed rather than by reviewed compared to all. This results in misleading population characteristics. For example, 47% of reviewed patients in group I were bitten by a Russell’s viper and 34% of all patients were bitten by a Russell’s viper. However (when I did the math), 28% of non-reviewed study subjects were bitten by a Russell’s viper compared to 47% of those reviewed. This is the more appropriate way to present the study. My strong recommendation is to re-create Table 1.

• Was/is it possible to evaluate the same group at one year of follow-up and then two and so on? This would give you better control of potential confounding factors.

• If the focus of the study is to compare symptoms and adverse health-related issues four years vs. one year after snakebite, why didn’t you conduct that very analysis? Supplementary table 2 shows the regression analyses but they are within group and not between group (which would be the most informative). Without this type of analysis, you only have descriptive within group data that does not directly address the scope of your research question.

Reviewer #3: 

A major issue is the question whether the reviewed patients can be considered as representative sample of all patients. The numbers in table 1 show that there might be some doubt. A higher percentage received antivenom, and of these, a higher percentage showed acute adverse reactions. This is only briefly discussed on lines 393-397. It is unclear, however, what impact that could have on the result. 

Table 1, last row: The percentage should be relative to those which received antivenom

Supplemental tables: since the logistic regression analysis is described in detail in the methods section, I wonder Why the results are hidden in the supplement.

It is not appropriate to give OR estimates, p-values and confidence intervals with four digits after the decimal point. two is more than enough. 

Supp. table 3 looks sophisticated, however not too useful. It is not clear how “Type of snake”, apparently the most relevant predictor, has been formalized and which snake is associated with a higher risk. Z-value and p-value can be derived from each other, thus one of both is redundant. The equations with the estimated regression coefficients are not useful either. 

CONCLUSIONS

Reviewer #2: 

There are many methodological limitations of this study. For example, the participation rates were relatively low, the outcome is based on “perception” which may be subject to information bias, and no relative measures of association are reported in the abstract.

DISCUSSION

Reviewer #3:

The discussion is rather lengthy and should be streamlined.However, the limitations of the study and possible directions of biased results need more emphases

SUMMARY AND GENERAL COMMENTS

Reviewer #1: 

The manuscript by Waiddyanatha et al. “Long-term health effects perceived by snakebite patients in rural 2 Sri Lanka: a cohort Study” concerns a crucial issue, with strong impacts on daily activities and livelihood of concerned people, and on the economy of large rural areas, and yet very poorly addressed.

The authors report a high proportion (25 to 50%) of people bitten by a snake complaining of symptoms, which were in the minority physical (amputations, functional deficits, musculoskeletal disorders, unsightly sequelae). However, most disorders were psychological in nature – or physical but attributed to biting without clear evidence – and may be culturally dependent. Their real impact on their daily lives is difficult to assess although probably not critical.

My major concern – partly mentioned by the authors – concerns the poor representativeness of the sample of patients interviewed by the authors, both geographically and statistically (low percentage of examined patients from the cohorts).

The authors could suggest modifying the protocol to overcome these flaws (in addition to recommend surveys in other regions to corroborate their results) and make it possible to verify the hypothesis regarding the possible role of cultural factors in the sequelae not directly linked to the action of the venom. It could be useful to include patients who have suffered from other severe or shocking pathologies to look for this type of sequelae. While waiting for such investigations, the authors could seek other studies already carried out on this subject regarding pathologies other than snakebite envenomation.

Reviewer #2: 

Discussion and Overall Methodological Comments:

• You did not perform the necessary regression analyses to compare four year vs. one year differences. Furthermore, such analyses would need to account for potentially relevant confounding factors. For example, if you were to compare a self-reported health issue at four years vs. one year, you would need to statistically adjust for relevant factors, such as the severity and location of the bite, timing of treatment, etc.

A between group table that shows logistic regression data should be included.

• Also, what about effect modification? For example, did systemic envenoming or antivenom modify the reported associations? This is very important and such stratified analyses would be informative.

• Very few methodological and analytical limitations are discussed in your paper. You need to acknowledge and address these important issues. The word “bias” does not even show up in your paper even though your study is subject to information bias, recall bias, and selection bias, among other types of biases (e.g., diagnostic bias). In addition, you do not discuss confounding even though this is a significant concern as I mentioned above.

Identity of reviewers:

Reviewer #1: Yes: Jean-Philippe Chippaux

Reviewer #2: No

Reviewer #3: No
---

## [Decision Letter · Decision Letter 1]

7 Jul 2022

Dear Prof Isbister,

Thank you very much for submitting your manuscript "Long-term health effects perceived by snakebite patients in rural Sri Lanka: a cohort study" for consideration at PLOS Neglected Tropical Diseases. As with all papers reviewed by the journal, your manuscript was reviewed by members of the editorial board and by several independent reviewers. In light of the reviews (below this email), we would like to invite the resubmission of a significantly-revised version that takes into account the reviewers' comments. 

see attached file (methodological) re-review

We cannot make any decision about publication until we have seen the revised manuscript and your response to the reviewers' comments. Your revised manuscript is also likely to be sent to reviewers for further evaluation.

Sincerely,

Thomas Junghanss

Guest Editor

Joerg Blessmann

Deputy Editor

see attached file (methodological) re-review

Reviewer's Responses to Questions

**Key Review Criteria Required for Acceptance?**

**Methods**

-Are the objectives of the study clearly articulated with a clear testable hypothesis stated?

-Is the study design appropriate to address the stated objectives?

-Is the population clearly described and appropriate for the hypothesis being tested?

-Is the sample size sufficient to ensure adequate power to address the hypothesis being tested?

-Were correct statistical analysis used to support conclusions?

-Are there concerns about ethical or regulatory requirements being met?

Reviewer #3: see attached file

**Results**

-Does the analysis presented match the analysis plan?

-Are the results clearly and completely presented?

-Are the figures (Tables, Images) of sufficient quality for clarity?

Reviewer #3: see attached file

**Conclusions**

-Are the conclusions supported by the data presented?

-Are the limitations of analysis clearly described?

-Do the authors discuss how these data can be helpful to advance our understanding of the topic under study?

-Is public health relevance addressed?

Reviewer #3: see attached file

**Editorial and Data Presentation Modifications?**

Reviewer #3: see attached file

**Summary and General Comments**

Reviewer #3: see attached file

PLOS authors have the option to publish the peer review history of their article (what does this mean?). If published, this will include your full peer review and any attached files.

Reviewer #3: No
---

## [Decision Letter · Decision Letter 2]

6 Aug 2022

Dear Prof Isbister,

We are pleased to inform you that your manuscript 'Long-term health effects perceived by snakebite patients in rural Sri Lanka: a cohort study' has been provisionally accepted for publication in PLOS Neglected Tropical Diseases.

Best regards,

Thomas Junghanss

Guest Editor

Joerg Blessmann

Section Editor

Reviewer's Responses to Questions

**Key Review Criteria Required for Acceptance?**

**Methods**

-Are the objectives of the study clearly articulated with a clear testable hypothesis stated?

-Is the study design appropriate to address the stated objectives?

-Is the population clearly described and appropriate for the hypothesis being tested?

-Is the sample size sufficient to ensure adequate power to address the hypothesis being tested?

-Were correct statistical analysis used to support conclusions?

-Are there concerns about ethical or regulatory requirements being met?

Reviewer #3: The description has improved

**Results**

-Does the analysis presented match the analysis plan?

-Are the results clearly and completely presented?

-Are the figures (Tables, Images) of sufficient quality for clarity?

Reviewer #3: The result of the multivariable model is more convincing now. There is a small typo ein the MLR equation. It should be

- 1.281(Time since snake bite, 4 years) rather than - 1.281(Time since snake bite, 1 year)

**Conclusions**

-Are the conclusions supported by the data presented?

-Are the limitations of analysis clearly described?

-Do the authors discuss how these data can be helpful to advance our understanding of the topic under study?

-Is public health relevance addressed?

Reviewer #3: no further comments

**Editorial and Data Presentation Modifications?**

Reviewer #3: no further comments

**Summary and General Comments**

Reviewer #3: no further comments

PLOS authors have the option to publish the peer review history of their article (what does this mean?). If published, this will include your full peer review and any attached files.

Reviewer #3: No

---

## [Editor Report · Acceptance letter]

26 Aug 2022

Dear Prof Isbister,

We are delighted to inform you that your manuscript, "Long-term health effects perceived by snakebite patients in rural Sri Lanka: a cohort study," has been formally accepted for publication in PLOS Neglected Tropical Diseases.

Best regards,

Shaden Kamhawi

co-Editor-in-Chief

Paul Brindley

co-Editor-in-Chief
